# Role of Gln79 in Feedback Inhibition of the Yeast γ-Glutamyl Kinase by Proline

**DOI:** 10.3390/microorganisms9091902

**Published:** 2021-09-07

**Authors:** Akira Nishimura, Yurie Takasaki, Shota Isogai, Yoichi Toyokawa, Ryoya Tanahashi, Hiroshi Takagi

**Affiliations:** Division of Biological Science, Graduate School of Science and Technology, Nara Institute of Science and Technology, Nara 630-0192, Japan; nishimura@bs.naist.jp (A.N.); takasaki.yurie.tw6@bs.naist.jp (Y.T.); s-isogai@bs.naist.jp (S.I.); y-toyokawa@bs.naist.jp (Y.T.); tanahashi.ryoya.ti3@bs.naist.jp (R.T.)

**Keywords:** feedback inhibition, *γ*-glutamyl kinase, proline, *Saccharomyces cerevisiae*

## Abstract

Awamori, the traditional distilled alcoholic beverage of Okinawa, Japan, is brewed with the yeast *Saccharomyces cerevisiae*. During the distillation process after the fermentation, enormous quantities of distillation residues containing yeast cells must be disposed of, and this has recently been recognized as a major problem both environmentally and economically. Proline, a multifunctional amino acid, has the highest water retention capacity among amino acids. Therefore, distillation residues with large amounts of proline could be useful in cosmetics. Here, we isolated a yeast mutant with high levels of intracellular proline and found a missense mutation (Gln79His) on the *PRO1* gene encoding the γ-glutamyl kinase Pro1, a limiting enzyme in proline biosynthesis. The amino acid change of Gln79 to His in Pro1 resulted in desensitization to the proline-mediated feedback inhibition of GK activity, leading to the accumulation of proline in cells. Biochemical and in silico analyses showed that the amino acid residue at position 79 is involved in the stabilization of the proline binding pocket in Pro1 via a hydrogen-bonding network, which plays an important role in feedback inhibition. Our current study, therefore, proposed a possible mechanism underlying the feedback inhibition of γ-glutamyl kinase activity. This mechanism can be applied to construct proline-accumulating yeast strains to effectively utilize distillation residues.

## 1. Introduction

Awamori, a type of shochu, is a traditional distilled alcoholic beverage made from steamed rice in Okinawa, Japan. It has been brewed with both the fungus *Aspergillus luchuensis* and the yeast *Saccharomyces cerevisiae* for more than 600 years. Awamori is manufactured by the following process of multiple parallel fermentation, which means simultaneous saccharification and fermentation [1]: (i) To make *koji*, *A. luchuensis* is inoculated into steamed rice. Amylolytic enzymes of *A. luchuensis* degrade the starch in the rice grain into glucose. (ii) The produced glucose is next fermented to ethanol by *S. cerevisiae* in the *moromi*, including a fermented mash. This step is called the *moromi* fermentation step. (iii) In the last step, the *moromi*, containing about 20% ethanol, is distilled to develop a rich and strong flavor, and the distillate is awamori. Although the distillation process is essential for awamori brewing, it creates an enormous amount of distillation residues containing yeast cells, and the disposal of these residues has become a major problem both environmentally and economically. Thus, a practical utilization of distillation residues is required. Some companies have reportedly filtered the residues and sold the resulting liquid as a healthy drink rich in amino acids and citrate. However, considering the large number of residues, more ways to use them effectively should be proposed. We have noticed that the residues contain large amounts of yeast cells, and we have long studied ways to utilize residues in this regard.

Proline is a multifunctional amino acid in various organisms. In addition to being a proteogenic amino acid, proline functions as an oxidative stress protectant, an osmolyte, a protein-folding chaperone, and a membrane stabilizer [2,3,4,5]. More importantly, proline has the highest water solubility and retention capacity among amino acids [6]. Therefore, proline is known to function as a natural moisturizing factor that retains water on the human skin [7]. In addition, since proline can directly scavenge reactive oxygen species (superoxide, hydrogen peroxide, etc.) generated by ultraviolet light from the sun, proline also protects the skin from the sun’s radiation [8]. Thus, proline may be a good addictive in cosmetics such as lotions and creams.

In *S. cerevisiae*, proline is synthesized from glutamate by three cytoplasmic enzymes: the γ-glutamyl kinase (GK) Pro1, the γ-glutamyl phosphate reductase Pro2, and the Δ^1^-pyrroline-5-carboxylate reductase (P5C) Pro3 (Appendix A) [9,10]. Meanwhile, in the proline degradation pathway, proline is oxidized to P5C by the mitochondrial proline oxidase Put1. P5C is then converted into glutamate by the P5C dehydrogenase Put2 (Appendix A) [11,12]. The *PUT1* and *PUT2* genes are regulated by the transcriptional activator Put3 [13]. Although Put3 is constitutively bound to the *PUT1* and *PUT2* promoters, Put3 is maximally activated for the upregulation of *PUT1* and *PUT2* only in the presence of proline. In addition to the transcriptional regulation of proline metabolisms, excess proline directly inhibits the enzymatic activity of Pro1 in a concentration-dependent manner [14]. Unlike bacteria and plants, yeasts do not accumulate proline in cells. However, our previous study revealed that the Ile150Thr variant of Pro1 was less sensitive to negative feedback inhibition by proline. Thus, yeast cells accumulate proline by expressing the mutant *PRO1*^I150T^ gene encoding the Ile150Thr variant [14]. Intriguingly, yeast cells with proline accumulation exhibited increased ethanol stress tolerance and fermentation ability [15]. Accordingly, yeast strains with high concentrations of proline may be appropriate for producing alcoholic beverages, such as sake and awamori.

GK is a highly conserved protein in many microorganisms and is the rate-limiting enzyme of proline biosynthesis. GK activity is strongly and allosterically inhibited by proline, and GK is known as a key enzyme regulating the intracellular proline content [16]. GK has been crystallized with glutamate, γ-glutamyl phosphate, and ADP, but the binding site of proline remains unknown. In crystals, γ-glutamyl phosphate is usually displaced to a 5-oxoproline molecule that is spontaneously cyclized from γ-glutamyl phosphate [17,18]. Surface analyses and docking calculations imply that the binding sites of glutamate partially overlap those of proline, indicating that proline binds to the substrate cavity [17]. However, the detailed mechanism underlying the proline-mediated feedback inhibition of GK is still unknown.

We now consider that a yeast mutant with high levels of proline will probably show two advantages in awamori brewing in the future. One is that it will enable the practical utilization of distillation residues containing high proline content for cosmetics. The other is that it will improve the fermentation ability of yeast. In this study, we first isolated the proline-accumulating awamori yeast mutant. We discovered that a missense mutation (Gln79His) on the *PRO1* gene occurs in the mutant, leading to desensitization to the proline-mediated feedback inhibition of GK activity. Additionally, our in silico analysis indicated that the amino acid residue at position 79 is involved in the stabilization of the proline binding pocket in Pro1 via a hydrogen-bonding network. Therefore, we propose a possible mechanism to explain the feedback inhibition of GK activity by proline.

## 2. Materials and Methods

### 2.1. Yeast Strains, Media, and Plasmids

A diploid awamori yeast strain 101-18 (supplied by National Research Institute of Brewing, Hiroshima, Japan) and a haploid laboratory yeast strain BY4741 (obtained from Euroscarf, Frankfurt, Germany) were used in this study.

A nutrient-rich medium YPD (1% yeast extract, 2% peptone, and 2% glucose) was used for random mutagenesis of 101-18 strain. A synthetic medium SD + Am (2% glucose, 0.5% ammonium sulfate, 0.67% yeast nitrogen base without ammonium sulfate and amino acids) was used for analysis of intracellular proline and glutamate contents in yeast. A synthetic medium SD + Alt (2% glucose, 0.5% allantoin, 0.67% yeast nitrogen base without ammonium sulfate and amino acids) was used for resistance tests of a proline analogue L-azetidine-2-carboxylic acid (AZC). All reagents were purchased from Sigma-Aldrich (St. Loius, MO, USA) unless otherwise stated.

Plasmids for the yeast expression of Pro1 variants (Q79A, Q79E, Q79H, Q79K, Q79N, Q79R, and Q79W) were constructed by the Quikchange method (Agilent, Santa Clara, CA, USA) with applicable primer sets (listed in Appendix A) and pAG416GPD-Pro1 WT [19] as a template. The yeast expression plasmids were introduced into strain BY4741 by the lithium acetate-PEG method [20]. Plasmids for the bacterial expression of Pro1 variants (Q79A, Q79E, Q79H, Q79K, Q79N, Q79R, Q79W, I150T, and D143A) were constructed by the Quikchange method (Agilent, Santa Clara, CA, USA) with applicable primer sets (listed in Appendix A) and pET53-Pro1 WT [19] as a template.

### 2.2. Isolation of AZC-Resistant Mutants from Awamori Yeast

Strain 101-18 was grown at 30 °C in YPD medium to the stationary growth phase and then treated with 5% ethyl methanesulfonate (EMS) in phosphate buffer (pH 7.0). After 60 min, 10% sodium thiosulfate was added to stop the mutagenesis reaction. The survival rate during mutagenesis was in the range of 10–20%. The cells were collected, washed twice with sterile water, and plated on SD + Alt containing 100 µg/mL AZC. After 3 days at 30 °C, many colonies were appeared as AZC-resistant mutants. Among them, we picked up one colony named as strain 18-Pro, which exhibited the highest intracellular proline content.

### 2.3. Spot Test

Yeast cells were grown at 30 °C in SD + Alt to the stationary growth phase and diluted to 1.0 of optical density at 600 nm (OD_600_) with water. Aliquots (3 µL) of 10-fold serial dilutions were spotted on SD + Alt in the absence or presence of AZC. The plates were then incubated at 30 °C for 3 days.

### 2.4. Quantification of Intracellular Proline and Glutamate Contents

Yeast cells were inoculated into SD + Am at an OD_600_ = 0.1. After incubation at 30 °C for 48 h under shaking, cells (40 of OD_600_ unit) were collected, resuspended with 1.0 mL of water, and subsequently boiled for 20 min to extract intracellular amino acids. After removal of cell debris by centrifugation for 5 min at 15,000× *g*, proline and glutamate content in the supernatant was quantified with an amino acid analyzer (JLC-500/V, JEOL, Tokyo, Japan) [21].

### 2.5. Homology Modeling of Pro1 and Docking Simulation of Proline into Pro1

The homology models of Pro1 were constructed using SWISS-model with a crystal structure (PDB ID: 2J5V) of the *Escherichia coli* ProB as a template [18]. ProB shows a 35% amino acid homology with Pro1. The docking of proline into the Pro1 model was carried out using SwissDock and default parameters.

### 2.6. Expression and Purification of Recombinant Pro1

Expression and purification of recombinant Pro1 were carried out as described previously [19] with modifications. pET53-Pro1 variants (Q79A, Q79E, Q79H, Q79K, Q79N, Q79R, Q79W, I150T, or D143A) were introduced into *E. coli* BL21 (DE3). *E coli* transformants were cultivated at 37 °C in M9CA medium (0.4% glucose, 2% casamino acid, 65 mM sodium phosphate, 8.6 mM NaCl, 18.7 mM ammonium chloride, 1 mM MgSO_4_, and 0.1 mM CaCl_2_) containing 100 µg/mL ampicillin to reach an OD_600_ of 0.8. Isopropyl *β*-D-1-thiogalactopyranoside was then added to a final concentration of 0.5 mM and further incubated for 20 h at 18 °C. Cells were harvested, suspended in 10 mL of buffer A (50 mM Tris-HCl (pH 7.4), 150 mM NaCl, and 20% (*w*/*v*) glycerol) and disrupted by ultrasonication under cooling. The supernatant was applied onto a nickel affinity column (Ni Sepharose 6 Fast flow, Cytiva, Tokyo, Japan). The column was washed with 100 mM imidazole in buffer A, and subsequently, purified proteins were eluted by buffer A containing 500 mM imidazole.

### 2.7. Assay of GK Activity of Pro1

GK activity was measured by the production of ADP in an enzyme-coupled system with pyruvate kinase (PK) and lactate dehydrogenase (LDH). The assay was carried out in 100 mM HEPES-NaOH pH 7.4, 400 mM sodium glutamate, 5 mM ATP, 10 mM MgCl_2_, 1 mM phosphoenoylpyruvate, 0.25 mM NADH, 7.5 U PK/LDH (Sigma-Adrich), and 1–8 μg of purified Pro1 in a total volume of 1 mL. The GK reaction mixture was pre-equilibrated for 3 min at 30 °C, and then sodium glutamate was added to initiate the enzymatic reaction. GK-dependent oxidation of NADH was monitored at 340 nm with a DU-800 spectrophotometer (Beckman Coulter, Brea, CA, USA) at 30 °C. In order to examine the feedback inhibition sensitivity, various concentration of proline (0–500 mM) was added to the reaction mixture. The reaction rate was calculated with the extinction coefficient of NADH, 6220 M^−1^ cm^−1^. One unit of activity was defined as the amount of enzyme required to produce 1 μmol of ADP per min.

### 2.8. Western Blotting

Samples were heat-denatured and separated via to SDS-polyacrylamide gel electrophoresis (10%), followed by transfer to polyvinylidene fluoride membranes (Cytiva). Membranes were blocked with Blocking One (Nacalai Tesque, Kyoto, Japan), after which they were incubated with antibodies in Can Get Signal Immunoreaction Enhancer Solution 1 (Toyobo, Shiga, Japan) at 4 °C overnight. Antibodies used in this study included the following: anti-hemagglutinin (HA)-probe (Y-11) (Santa Cruz Biotechnology, Santa Cruz, CA, USA) and anti-glyceraldehyde-3-phosphate dehydrogenase (GAPDH) (GA1R) (Thermo Fisher Scientific, Rockland, IL, USA). Membranes were washed three times in TTBS (20 mM Tris-HCl (pH 8.0), 0.1 M NaCl, and 0.02% (*v*/*v*) Tween-20) and then incubated with a horseradish peroxidase-conjugated secondary antibody in Can Get Signal Immunoreaction Enhancer Solution 2 (Toyobo) for 1 h at room temperature. After the membranes were washed again three times in TTBS, immunoreactive bands were detected via a chemiluminescence reagent (ECL Prime Western Blotting Detection Reagent; Cytiva) with a luminescent image analyzer (ImageQuant LAS 4000, Cytiva).

### 2.9. Statistical Analysis

Results are presented as means ± standard deviations (SD) of three experiments. Student’s *t*-tests for two-group comparisons and one-way/two-way ANOVAs with Tukey’s tests for multiple-group comparisons were performed to evaluate statistical significance using GraphPad Prism 7 (GraphPad Software, San Diego, CA, USA). *p* < 0.05 was considered to be statistically significant.

## 3. Results and Discussion

### 3.1. Isolation of Awamori Yeast with Intracellular Proline Accumulation

We believe that the accumulation of proline in awamori yeast cells is beneficial for the practical use of distillation residues. When random mutagenesis, such as treatment with EMS, is introduced into the diploid industrial strain, such as awamori yeast, the resulting mutants carry many heterozygous mutations and a few homozygous mutations in the genome. In the case of a recessive mutation, the wild-type gene remaining in one genome masks the effect of heterozygous mutation in another genome, indicating that homozygous mutation is required for expressing the favorable phenotype. The low frequency of homozygous mutation will be difficult to obtain recessive mutants, such as auxotrophy. Moreover, many of the diploid industrial yeast strains, such as sake and baker’s yeasts, could poorly sporulate, resulting in low efficiency to form haploids, which limits the ability to obtain desirable diploid strains by mating haploid mutants. In contrast, a dominant mutation can alter the phenotypes even in the case of heterozygous mutation. In general, mutations with enhanced amino acid biosynthesis are dominant, mainly due to the removal of feedback inhibition of the key enzyme, such as Pro1, implying that an awamori yeast mutant with proline accumulation can be obtained by conventional mutagenesis. Here, we attempted to use a proline toxic analogue, AZC, to construct proline-accumulating mutants. AZC can compete with proline for incorporation into nascent proteins, resulting in cell death [22]. By a random mutagenesis with EMS, we isolated an AZC-resistant mutant strain, 18-Pro, from the diploid awamori yeast strain 101-18 (Figure 1a). Next, we determined the intracellular proline content in both parent strain 101-18 and mutant strain 18-Pro. As we expected, the intracellular level of proline in 18-Pro was much higher than that in 101-18 (10.5 vs. 1.1 nmol/OD_600_ unit) (Figure 1b). It has been reported that reduced sensitivity to proline-mediated feedback inhibition of GK causes the accumulation of proline in the cell [14,17]. Therefore, we sequenced the *PRO1* gene encoding GK in strains 101-18 and 18-Pro. The results revealed that 18-Pro has a mixture of nucleotides G and C at position 237, whereas 101-18 only has a nucleotide G at the same position. The mutation of G to C leads to the amino acid replacement of Gln to His at position 79 (Q79H), suggesting that 18-Pro has a heterozygous missense mutant of *PRO1*. Homology analysis of the Pro1 sequence indicated that Gln79 is completely conserved in the Pro1 homologue of other microorganisms, suggesting the importance of Gln at position 79 (Appendix A). Our previous study showed that amino acid residues around position 150 in Pro1 are important for the mechanism underlying feedback inhibition by proline [14]. The replaced position (Gln79) found in this study seems to be away from position 150 based on the primary structure. However, the homology model implied that both residues at positions 79 and 150 are surrounded by the active site binding γ-glutamyl phosphate or 5-oxoproline, which are the enzymatic product and its derivative of GK, respectively (Figure 1c).

### 3.2. Characterization of the Q79H Variant of Pro1

To confirm the effect of the Q79H substitution on the accumulation of proline, we expressed the Q79H variant of Pro1 in a laboratory yeast strain, BY4741. As shown in Figure 2a, yeast cells expressing Q79H showed a higher tolerance to AZC than the cells expressing WT. More importantly, Figure 2b shows that the expression of the Q79H variant contributed to the increase in intracellular proline content. No significant difference was found in the intracellular proline content between the empty vector control and WT expression, indicating that the overexpression of wild-type Pro1 does not increase the intracellular proline content, which is consistent with a previous report [23]. We further analyzed the GK activity of WT Pro1 and variants (Q79H and I150T) using recombinant proteins to investigate the feedback inhibition sensitivity by proline (Appendix A). The I150T variant of Pro1 is known to be insensitive to feedback inhibition. The specific activity of GK (18.3 U/mg) in the Q79H variant was similar to that in WT (17.4 U/mg) and to that in I150T (21.0 U/mg) in the absence of proline. In the presence of proline, the GK activity in WT was dramatically inhibited and completely disappeared in 10 mM proline (Figure 3). It should be noted that the relative activity in the Q79H variant was about 96% even in the presence of 500 mM proline, which was much higher than that in the I150T variant (50%) (Figure 3). These results showed that the Q79H variant is less sensitive to feedback inhibition by proline, leading to proline accumulation in yeast cells.

### 3.3. Importance of Amino Acid Residue at Position 79 within Pro1

The importance of Gln79 in the mechanism underlying the feedback inhibition of GK is still unknown. Therefore, we prepared recombinant Pro1 variants (Q79A, Q79E, Q79K, Q79N, Q79R, and Q79W) that replaced Gln79, with some amino acids having similar or different properties (Appendix A). We first measured the enzyme activity of GK under conditions without proline (Table 1). The specific activities in Q79A, Q79N, and Q79W were almost the same as that in WT. On the other hand, the specific activity in Q79E and Q79R was lower than that in WT. Especially, the activity in Q79E was dramatically reduced. Next, we measured the GK activity in Pro1 variants in reaction solutions with various concentrations of proline (Figure 4). The activities in Q79R and Q79W were observed even in the presence of 500 mM proline. In particular, Q79R showed almost the same activity as Q79H in the reaction containing 500 mM proline. The sensitivity of feedback inhibition in Q79K and Q79N was decently reduced compared to that in WT. The activities in Q79A and Q79E were slightly higher than that in WT in the presence of proline. These results indicated that the specific activity and sensitivity to feedback inhibition were both significantly changed by the substitution of amino acid residue at position 79, suggesting that an amino acid residue at position 79 is important for the regulation of enzyme activity.

The detailed mechanism by which an amino acid change at position 79, except Gln, reduces the sensitivity to feedback inhibition by proline is unclear. Since GK uses glutamate, an acidic amino acid, as a substrate, Q79R or Q79K, in which Gln was changed to a basic amino acid residue, might recruit glutamate in the catalytic pocket. Therefore, the affinity for the competitive inhibitor, proline, might be relatively reduced. On the other hand, when the amino acid residue at position 79 is changed to acidic amino acid Glu, glutamate can hardly access the catalytic pocket due to charge repulsion, resulting in reduced activity. Feedback sensitivity by proline was also dramatically reduced in the Q79W variant. Since proline is the most hydrophilic amino acid, the hydrophobic side chain of Trp is likely to eliminate the entry of proline into the catalytic pocket.

To investigate whether the replacement of Gln79 with some amino acids affects the intracellular proline content in yeast, we expressed Pro1 variants into the laboratory yeast strain BY4741. As shown in Figure 5a, there was no difference in the Pro1 protein levels among yeast cells expressing each Pro1 variant (Figure 5a). We next spot tested each strain on the AZC-containing medium (Figure 5b). As shown above, the WT-expressing strain could not grow on the AZC-containing medium at all, while the Q79H-expressing strain clearly grew even in the presence of AZC. The I150T-expressing strain also showed obvious growth on the AZC-containing medium, as previously reported [14]. Strains that expressed variants other than Q79A and Q79E (Q79K, Q79N, Q79R, and Q79W) grew on the AZC-containing medium. In contrast, the Q79A- and Q79E-expressing strains did not grow in the presence of AZC. We also determined the intracellular proline content in Pro1 variant-expressing strains (Figure 5c). As expected, the strains expressing Q79H and I150T variants had significantly increased proline contents compared to WT. In addition, strains expressing Q79K, Q79N, Q79R, and Q79W also showed significant increases in intracellular proline contents compared to WT. On the other hand, there was no significant difference in the intracellular proline content among the strains expressing Q79A, Q79E, and WT. In yeast, proline is synthesized from glutamate, the substrate of Pro1 [14]. Thus, we measured the glutamate content in yeast cells (Appendix A). The strains expressing Q79H and I150T contained significantly less intracellular glutamate than the WT-expressing strain, indicating that the increased proline was biosynthesized via the Pro1-mediated pathway. It should be noted that although yeast transformant cells possess the wild-type *PRO1* gene in the chromosome and the *PRO1* mutant gene in the plasmid, the *PRO1* mutation involved in the removal of the feedback inhibition of Pro1 is dominant. Therefore, the feedback inhibition-insensitive Pro1 variants confer proline accumulation on BY4741 cells (Figure 2b and Figure 5c).

It has become apparent that excess proline has negative effects on cell growth or protein functions in yeasts and plants [24]. In addition, excessive enhancement of proline synthesis via phosphorylation of glutamate by Pro1 will decrease not only glutamate, which is the substrate for Pro1, but also intracellular redox cofactors and ATP, which are necessary for proline biosynthesis, leading to the metabolic imbalance in yeast cells. For the application of proline-accumulating strain, consumers and the food industry in Japan have not accepted genetically-modified (GM) yeast strains yet; however, it has potential to be utilized for the production of proline as materials for cosmetics. In the case of non-GM yeast, proline-accumulating strains with enhanced stress tolerance are expected to contribute to the food industry for efficient production of breads and alcoholic beverages [25,26].

We searched for amino acid residues that interact with Gln79 using the structure model of Pro1 (Figure 1c) to elucidate the role of amino acid residue at position 79. Figure 6a shows that Asp143 and Ser146 interact with Gln79 via a hydrogen bond. Both Asp143 and Ser146 were certainly conserved among various microorganisms (Appendix A). In particular, Asp143 was highly conserved. Interestingly, the hydrogen-bonding network, consisting of Gln79-Asp143-Ser146, was not observed in the model with the Q79H variant of Pro1, indicating the importance of Asp143 and Ser146 in the feedback inhibition by proline (Figure 6b). Incidentally, our previous report indicated that the replacement of Ser146 with Pro increases the proline content in yeast cells [27]. Therefore, we prepared the D143A variant of Pro1 (Appendix A). The specific activity (0.2 U/mg) of D143A was significantly lower than that of WT (17.4 U/mg). We next measured the GK activity of D143A in reaction solution containing proline (Figure 7). Interestingly, the D143A variant showed residual high activity of about 95% even in the presence of a high concentration (500 mM) of proline, indicating that the feedback inhibition was almost eliminated. These results indicated that Asp143 has an important role in both GK activity and proline-mediated feedback inhibition. Finally, a docking simulation of proline into Pro1 was performed using the structure model of Pro1 (Figure 8). The docking model suggested that Asp143 is the direct binding site of proline, raising the possibility that the position of Asp143 is essential for feedback inhibition by proline. Taken together, the present results indicate that Gln79-Asp143-Ser146 probably stabilizes the structure of the proline-binding pocket. In other words, when the amino acid residue at position 79 is changed to His from Gln, the Gln79-Asp143-Ser146 hydrogen-bonding network is collapsed, leading to a decrease in the affinity for proline.

## 4. Summary

In this study, we isolated a yeast mutant with high levels of intracellular proline and found a missense mutation (Gln79His) on the *PRO1* gene. The Gln79His substitution resulted in desensitization to the proline-mediated feedback inhibition of GK activity, leading to the high accumulation of proline in cells. In addition, the amino acid residue at position 79 plays an important role in the mechanism underlying feedback inhibition. The present study, therefore, demonstrates a possible mechanism underlying the feedback inhibition of GK activity. This mechanism can be applied to construct proline-accumulating yeast strains that can effectively utilize distillation residues.

## Figures and Tables

**Figure 1 microorganisms-09-01902-f001:**
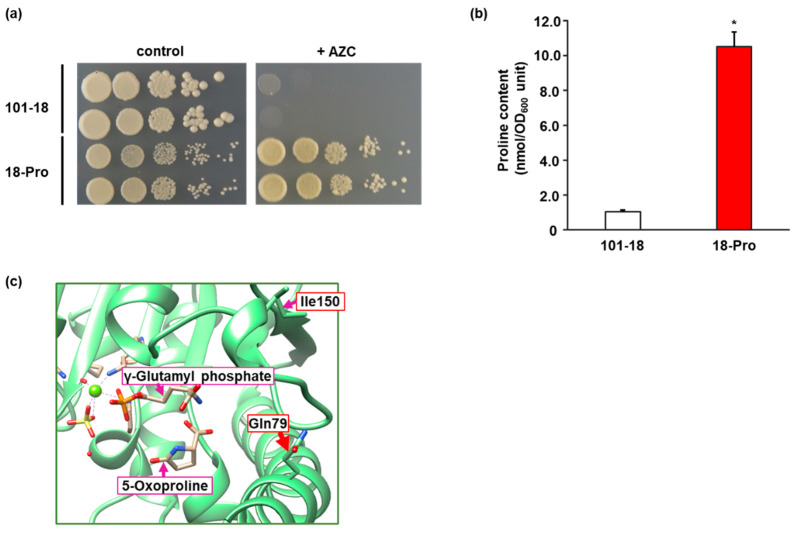
Isolation of awamori yeast mutant with intracellular proline accumulation. (**a**) Growth of strains 101-18 and 18-Pro on the AZC-containing medium. Yeast cells from each strain were spotted onto SD + Alt medium plate in the absence (control) and presence (+AZC) of AZC (100 µg/mL). The plates were incubated at 30 °C for 2–3 days. (**b**) Intracellular proline contents. Strains 101-18 and 18-Pro were grown on SD + Am medium and intracellular proline contents were determined. Data are presented as means ± SD (*n* = 3) and statistical significance was determined by Student *t*-test. * *p* < 0.05. (**c**) Homology model of Pro1. The homology modeling of Pro1 was carried out using Swiss-Model repository with a crystal structure (PDB ID: 2J5V) of the *E. coli* ProB. γ-Glutamyl phosphate is an enzymatic reaction product of Pro1 and binds to a position similar to that of the substrate glutamate. 5-oxoproline, which is a nonenzymatically cyclized form of γ-glutamyl phosphate, is bound to a position similar to that of proline. Ile150 and Gln79 indicate isoleucine at position 150 and glutamine at position 79 in Pro1, respectively.

**Figure 2 microorganisms-09-01902-f002:**
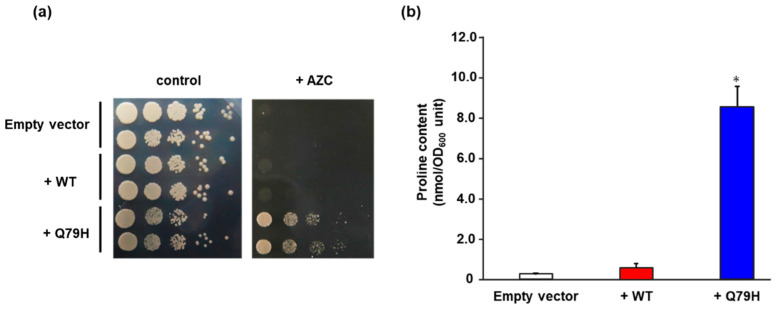
Characterization of the Q79H variant of Pro1. (**a**) Growth of yeast cells expressing the WT and Q79H Pro1 on the AZC-containing medium. The WT (+WT) and Q79H (+Q79H) Pro1 were expressed in the laboratory strain BY4741 and yeast cells from each strain were spotted onto an SD + Alt medium plate in the absence (control) and presence (+AZC) of AZC (250 µg/mL). Empty vector indicates a negative control strain. The plates were incubated at 30 °C for 2–3 days. (**b**) Intracellular proline contents. Yeast cells expressing the WT (+WT) and Q79H (+Q79H) Pro1 were grown on SD + Am medium and intracellular proline contents were determined. Empty vector indicates a negative control strain. Data are presented as means ± SD and statistical significance was determined by a one-way ANOVA with Tukey’s test. * *p* < 0.05, vs. WT.

**Figure 3 microorganisms-09-01902-f003:**
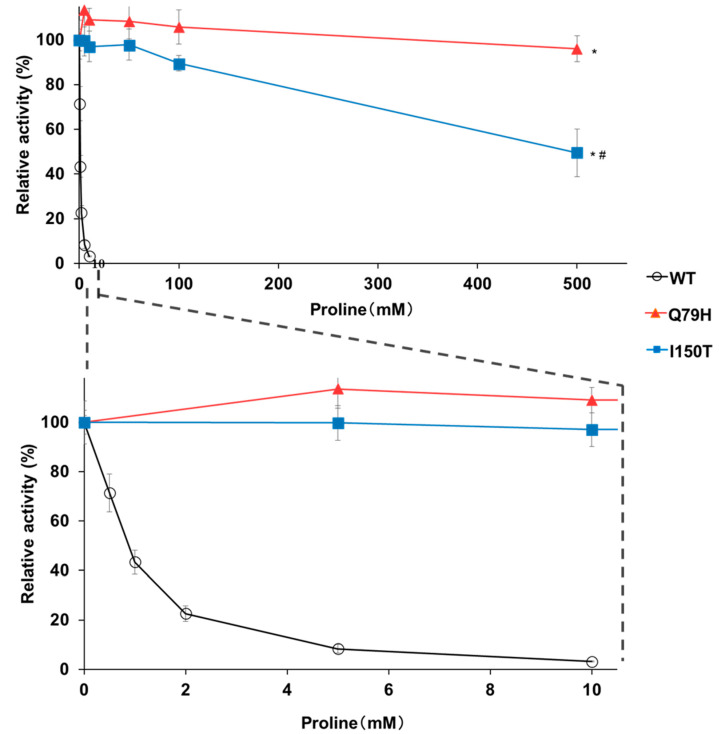
Feedback inhibition of GK activity of the Q79H variant by proline. The relative GK activities of the recombinant WT and variants (Q79H and I150T) Pro1 were measured in the presence of proline. The enzymatic activities in the absence of proline are defined as 100%. The upper panel represents the relative activities in the presence of 0–500 mM proline. The lower panel is a zoom of the upper panel at a concentration of 0–10 mM proline. Data are presented as means ± SD (*n* = 3) and statistical significance was determined by a two-way ANOVA with Tukey’s test. * *p* < 0.05, vs. WT; # *p* < 0.05, vs. Q79H.

**Figure 4 microorganisms-09-01902-f004:**
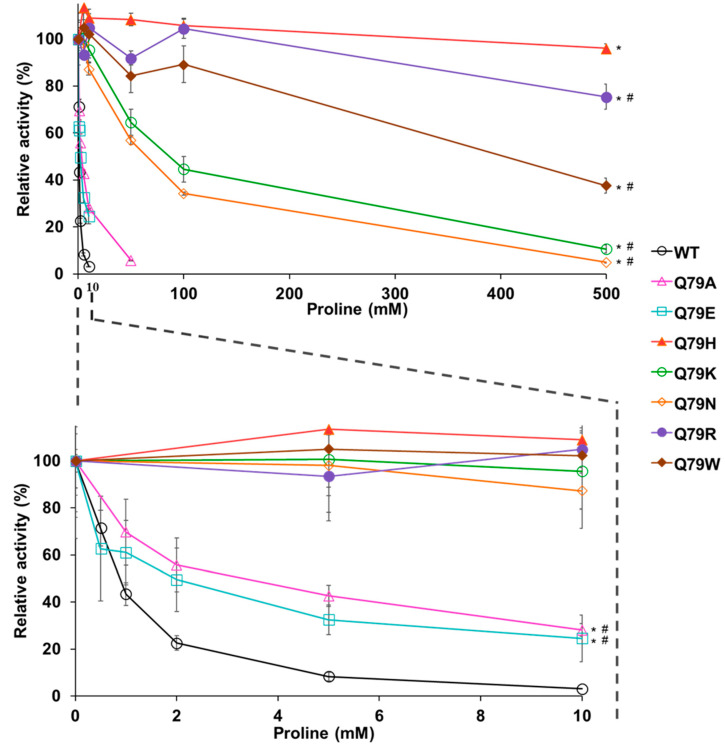
Effect of amino acid change at position 79 in Pro1 on feedback inhibition of GK by proline. The relative GK activities of the recombinant WT, Q79A, Q79E, Q79H, Q79K, Q79N, Q79R, and Q79W variant Pro1 were measured in the presence of proline. The enzymatic activities in the absence of proline are defined as 100%. Upper panel represents relative activities in the presence of 0–500 mM proline. The lower panel is a zoom of the upper panel at a concentration of 0–10 mM proline. Data are presented as means ± SD (*n* = 3) and the statistical significance was determined by a two-way ANOVA with Tukey’s test. * *p* < 0.05, vs. WT; # *p* < 0.05, vs. Q79H.

**Figure 5 microorganisms-09-01902-f005:**
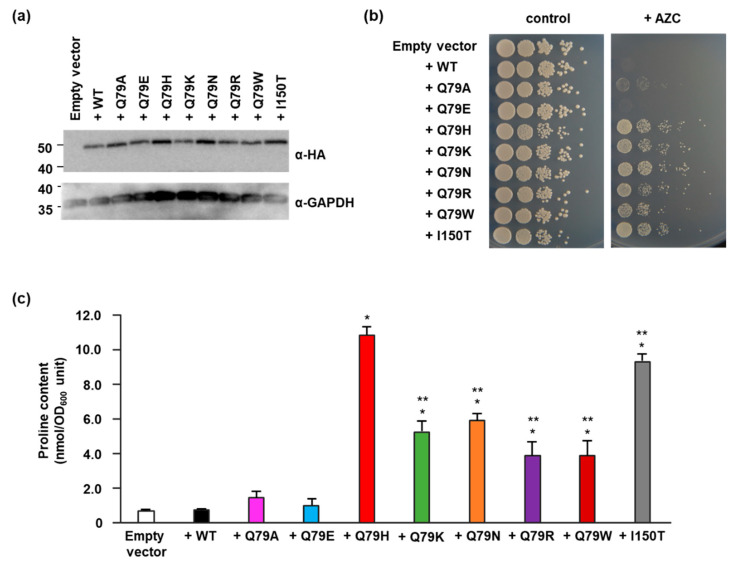
Effect of the Gln79 variants of Pro1 on proline accumulation in yeast cells. (**a**) Protein levels of Pro1 in yeast cells. Wild-type (+WT), Q79A (+Q79A), Q79E (+Q79E), Q79H (+Q79H), Q79K (+Q79K), Q79N (+Q79N), Q79R (+Q79R), or Q79W (+Q79W) variant Pro1 that were expressed in laboratory strain BY4741 and hemagglutinin (HA)-tagged Pro1 and glyceraldehyde-3-phosphate dehydrogenase (GAPDH) were detected with HA antibody (α-HA) and GAPDH antibody (α-GAPDH), respectively. GAPDH is shown as a loading control. Five µg of protein obtained from whole cell lysate was loaded. Uncropped western blot images are shown in Appendix A. (**b**) Growth of yeast cells expressing the Gln79 variants of Pro1 on the AZC-containing medium. Wild-type (+WT), Q79A (+Q79A), Q79E (+Q79E), Q79H (+Q79H), Q79K (+Q79K), Q79N (+Q79N), Q79R (+Q79R), or Q79W (+Q79W) variant Pro1 were expressed in laboratory strain BY4741 and each strain was spotted on SD + Alt medium plate in the absence (control) and presence (+AZC) of AZC (250 µg/mL). The plates were incubated at 30 °C for 2–3 days. (**c**) Proline contents in yeast cells expressing the Gln79 variants of Pro1. Wild-type (+WT), Q79A (+Q79A), Q79E (+Q79E), Q79H (+Q79H), Q79K (+Q79K), Q79N (+Q79N), Q79R (+Q79R), or Q79W (+Q79W) variant Pro1 were expressed in laboratory strain BY4741. Each strain was grown on SD + Am medium and intracellular proline content in each strain was determined. Empty vector indicates a negative control. Data are presented as means ± SD and statistical significance was determined by a one-way ANOVA with Tukey’s test. * *p* < 0.05, vs. WT; ** *p* < 0.05, vs. Q79H.

**Figure 6 microorganisms-09-01902-f006:**
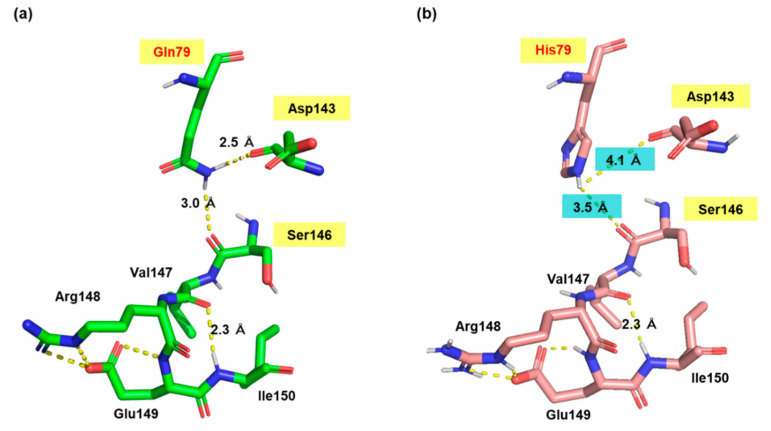
Comparison of local structure around position 79 in Pro1. Homology models of WT (**a**) and Q79H (**b**) Pro1 were constructed using SWISS-model with the crystal structure (PDB ID: 2J5V) of the *E. coli* ProB as a template. Local structures around position 79 in Pro1 are shown in the figures.

**Figure 7 microorganisms-09-01902-f007:**
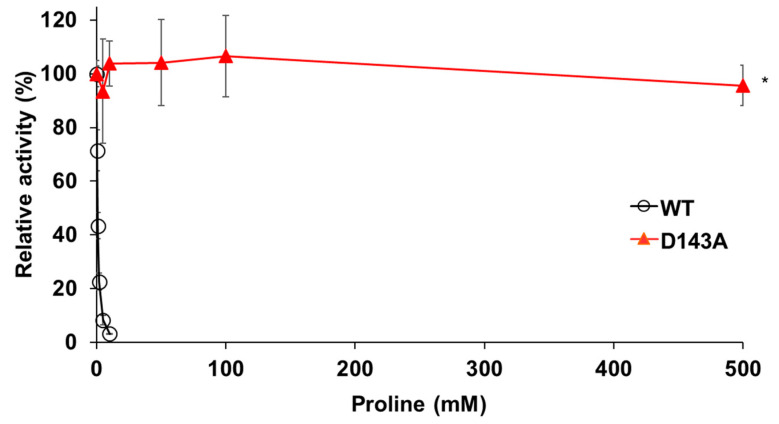
Effect of Asp-to-Ala substitution at position 143 in Pro1 on feedback inhibition of GK by proline. The relative GK activities of recombinant WT and D143A variant of Pro1 were measured in the presence of proline. The GK activities in the absence of proline are defined as 100%. Data are presented as means ± SD (*n* = 3) and statistical significance was determined by a two-way ANOVA with Tukey’s test. * *p* < 0.05, vs. WT.

**Figure 8 microorganisms-09-01902-f008:**
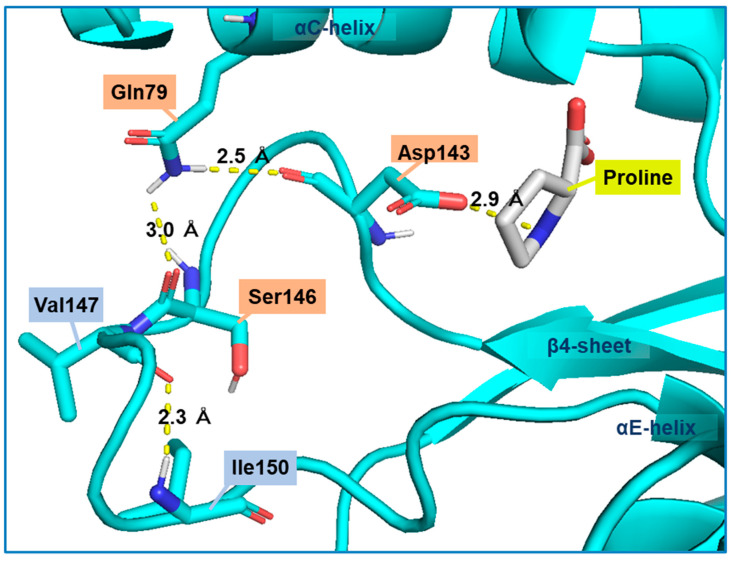
Possible model for stabilization of the proline-binding pocket via a hydrogen-bonding network among Gln79, Asp143, and Ser146. A homology model of WT Pro1 was constructed using SWISS-model with the crystal structure (PDB ID: 2J5V) of the *E. coli* ProB as a template. The docking of proline into the Pro1 model was carried out using SwissDock and default parameters.

**Table 1 microorganisms-09-01902-t001:** GK activities of the recombinant Pro1 proteins.

	WT	Q79A	Q79E	Q79K	Q79N	Q79R	Q79W
Specific activity (U/mg)	17.38 ± 0.84	17.88 ± 3.88	2.18 ± 0.72	13.58 ± 1.97	19.13 ± 2.20	9.12 ± 1.04	18.36 ± 4.39

## Data Availability

The analyzed data presented in this study are included within this article. Further data are available on reasonable request from the corresponding author.

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
