# Peer review of "Role of Gln79 in Feedback Inhibition of the Yeast γ-Glutamyl Kinase by Proline"

_microorganisms, 2021, doi:10.3390/microorganisms9091902_

Round 1
Reviewer 1 Report
Line 84, 88, 96...: "an awamori yeast mutant" It is better delete awamori in this case
Line 105: AZC is not described
Line 118: " the colony was picked" authors obtained only one colony after the mutagenesis?
Line 127: "40 of OD600 unit" is this correct?
Line 221: The expression of Q79H variant of Pro1 in a laboratory yeast strain, BY4741 , is not clear. Did the authors transform the plasmid into BY strain? Did the By strain present the WT PRO1? What was the transformation method? I suppose that authors have used the pro1 mutant of BY4741 collection, but this is not specified in the text and if is not a mutant the results would need a reinterpretation
Figure 3 and 4. authors should explain that the second panel is a zoom of the first one
Line 253: it is not well explained how the variant are constructed and expressed, E.coli? yeast?
Figure 5a. How much protein was loaded onto the gel?
Author Response
Reviewer 1:
We are very grateful to your beneficial and critical comments to our manuscript. According to your suggestion, we have made the revision carefully and appropriately.
Line 84, 88, 96...: "an awamori yeast mutant" It is better delete awamori in this case.
We have deleted “awamori” from the sentence (Line 84), but we prefer to retain the use of “awamori” yeast because we first isolated the proline-accumulating mutant from strain 101-18, which is officially named as “awamori yeast” (Lines 88 and 96). I would really appreciate it if you would kindly understand our response.
Line 105: AZC is not described
This is our careless mistake. We have added a description of AZC (Line 105).
Line 118: "the colony was picked" authors obtained only one colony after the mutagenesis?
Actually, many colonies were appeared as AZC-resistant mutants. Among them, we picked up one colony named as strain 18-Pro, which exhibited the highest intracellular proline content (Lines 120-122)
Line 127: "40 of OD600 unit" is this correct?
One OD600 unit is defined as cells in one milliliter of cell suspension at OD600 of 1.0. Therefore, “40 of OD600 unit” means cells in the one milliliter of cell suspension at OD600 of 40.
Line 221: The expression of Q79H variant of Pro1 in a laboratory yeast strain, BY4741, is not clear. Did the authors transform the plasmid into BY strain? Did the BY strain present the WT PRO1? What was the transformation method? I suppose that authors have used the pro1 mutant of BY4741 collection, but this is not specified in the text and if is not a mutant the results would need a reinterpretation.
For the expression of the Pro1 variant in yeast, we constructed the expression plasmids as described in the text (Lines 106-109). Subsequently, these episomal plasmids were introduced into strain BY4741 by the lithium acetate-PEG method [20] (Lines 109-110). Although yeast transformant cells possess the wild-type PRO1 gene in the chromosome and the PRO1 mutant gene in the plasmid, the PRO1 mutation involved in the removal of the feedback inhibition of the γ-glutamyl kinase Pro1 is dominant. Therefore, the feedback inhibition-insensitive Pro1 variants confers proline accumulation on BY4741 cells (Figures 2b and 5c) (Lines 343-346).
Figure 3 and 4. authors should explain that the second panel is a zoom of the first one
According to your suggestion, we have added the explanation of two graphs to the legends of Figure 3 and 4 (Lines 280-281; 319-320).
Line 253: it is not well explained how the variant are constructed and expressed, E.coli? yeast?
When we analyzed the effect of the expression of the mutated PRO1 genes on the Pro1 protein level and proline content in yeast cells, we expressed the PRO1 genes in strain BY4741 by episomal plasmids as described in the text (Lines 106-110). On the other hand, in order to analyze γ-glutamyl kinase activity of the Pro1 variants, we expressed and purified the recombinant Pro1 proteins using E. coli cells (Lines 110-113; 140-153).
Figure 5a. How much protein was loaded onto the gel?
We loaded 5 mg of protein obtained from whole cell lysate for western blotting. This explanation has been included in the legend of Figure 5a (Line 367).

Reviewer 2 Report
Please graphically present the biosynthetic pathway of proline with emphasize on steps from glutamate.
Please compare the advantages of using mutated PRO1 gene as opposed to overexpression of PRO1 gene in terms of efficiency.
Please elaborate on the downsides of intracellular accumulation of proline and application of (GMO) proline producing strain?
Please elaborate on complexity of setting up the mutated gene onto the diploid industrial strain.
Thanks!
Author Response
Reviewer 2:
We are very grateful to your beneficial and critical comments to our manuscript. According to your suggestion, we have made the revision carefully and appropriately.
Please graphically present the biosynthetic pathway of proline with emphasize on steps from glutamate.
According to your suggestion, we have added the metabolic pathway of proline in S. cerevisiae (Lines 59, 61, 438-439) as Figure S1 in Supplementary Materials.
Please compare the advantages of using mutated PRO1 gene as opposed to overexpression of PRO1 gene in terms of efficiency.
The wild-type Pro1 is highly sensitive to feedback inhibition by proline (Figure 3); therefore, the overexpression of the wild-type PRO1 gene did not increase the intracellular proline content (Figures 2b and 5c). We have also reported that proline content was virtually unchanged in S. cerevisiae cells which overexpress the wild-type Pro1 [23]. On the other hand, the PRO1 mutation involved in the removal of feedback inhibition of the γ-glutamyl kinase Pro1 is dominant. Therefore, the feedback inhibition-insensitive Pro1 variants confers proline accumulation on BY4741 transformant cells (Figures 2b and 5c) (Lines 343-346).Thus, the expression of the mutated PRO1 gene has great advantage for high production of proline in yeast cells compared with the overexpression of the wild-type PRO1 gene.
Please elaborate on the downsides of intracellular accumulation of proline and application of (GMO) proline producing strain?
According to your suggestion, we have included the below descriptions in the revised manuscript (Lines 348-357). It has become apparent that ‘excess’ proline has negative effects on cell growth or protein functions in yeasts and plants [24]. In addition, Pro1 variants will catalyze the phosphorylation of glutamate to γ-glutamyl phosphate even in the presence of proline due to its desensitization of feedback inhibition by proline. Consequently, intracellular content of redox cofactors and ATP, which are necessary for the proline biosynthesis, and glutamate, which is the substrate for Pro1, will be decreased, leading to the metabolic imbalance in yeast cells. For the application of proline-accumulating strain, consumers and food industry in Japan have not accepted genetically-modified (GM) yeast strain yet; however, it has potential to be utilized for high production of bioethanol or materials for cosmetics. In the case of non-GM yeast, proline-accumulating strains with enhanced stress tolerance are expected to contribute to the food industry for efficient production of breads and alcoholic beverages.
Please elaborate on complexity of setting up the mutated gene onto the diploid industrial strain.
Following your suggestion, we have included the below descriptions in the revised manuscript (Lines 191-204). When random mutagenesis, such as treatment with EMS, is introduced into the diploid industrial strain like awamori yeast, the resulting mutants carry many heterozygous mutations and a few homozygous mutations in the genome. In the case of a recessive mutation, the wild-type gene remaining in one genome masks the effect of heterozygous mutation in another genome, indicating that homozygous mutation is required for expressing the favorable phenotype. The low frequency of homozygous mutation will be difficult to obtain recessive mutants, such as auxotrophy. Moreover, many of the diploid industrial yeast strains, such as sake and baker’s yeasts, could poorly sporulate, resulting in low efficiency to form haploids, which limits to obtain desirable diploid strains by mating haploid mutants. In contrast, a dominant mutation can alter the phenotypes even in the case of heterozygous mutation. In general, mutations with enhanced amino acid biosynthesis are dominant, mainly due to the removal of feedback inhibition of the key enzyme, such as Pro1, implying that awamori yeast mutant with proline accumulation can be obtained by conventional mutagenesis.
